# Compressive Performance of Longitudinal Steel-FRP Composite Bars in Concrete Cylinders Confined by Different Type of FRP Composites

**DOI:** 10.3390/polym15204051

**Published:** 2023-10-11

**Authors:** Maojun Duan, Yu Tang, Yusheng Wang, Yang Wei, Jiaqing Wang

**Affiliations:** College of Civil Engineering, Nanjing Forestry University, Nanjing 210037, China; dmj@njfu.edu.cn (M.D.); 8210610637@njfu.edu.cn (Y.W.); wy78@njfu.edu.cn (Y.W.); jiaqingw@njfu.edu.cn (J.W.)

**Keywords:** compressive behavior, steel-fiber-reinforced composite bar, concrete cylinder, fiber-reinforced polymer confinement

## Abstract

This paper presents an experimental study on the compressive performance of longitudinal steel-fiber-reinforced polymer composite bars (SFCBs) in concrete cylinders confined by different type of fiber-reinforced polymer (FRP) composites. Three types of concrete cylinders reinforced with (or without) longitudinal SFCBs and different transverse FRP confinements were tested under monotonic compression. The results showed that the post-yield stiffness of SFCBs is higher when confined with high elastic modulus carbon fiber-reinforced polymer (CFRP) composite than with low elastic modulus basalt fiber-reinforced polymer (BFRP) composite. Decreasing confinement spacing did not significantly improve the compressive strength of SFCBs in concrete cylinders. The compressive failure strain of SFCBs could possibly reach 88% of its tensile peak strain in concrete cylinders confined by CFRP sheets, which is significantly higher than the value (around 50%) in previous studies. Existing design equations, which applied a strength reduction factor or a maximum compressive strain of concrete to consider the compressive contributions of SFCBs in concrete members, underestimate the load-carrying capacity of SFCB-reinforced concrete cylinders. The design equation that considers the actual compressive stress of SFCBs gives the most accurate prediction; however, its applicability and accuracy need to be verified with more experimental data.

## 1. Introduction

Modern performance-based seismic design (PBSD) requires structures to be repairable after certain levels of earthquake [1,2,3]. However, traditional steel-reinforced concrete (RC) structures usually experience uncontrollable damage during earthquakes due to the elastoplastic characteristics of steel reinforcement [4,5]. Using a deformation-controllable reinforcement appears to be an effective way to reduce the overall damage of RC structures after an earthquake, while their existing design method could be applied [6,7,8,9]. Wu et al. [10] proposed a steel-fiber-reinforced composite bar (SFCB), which is pultruded with an inner core steel bar and an outer fiber-reinforced polymer (FRP) layer, as shown in Figure 1a. The outer longitudinal fibers wrap the inner steel bar, providing superior anti-corrosion ability [11,12] and good interface bonding performance [13,14]. The composite effect of the outer elastic FRP layer and inner elastoplastic steel bar gives the SFCB a stable positive post-yield stiffness under both tensile and compressive loading [15], as shown in Figure 2. This significantly reduces the residual displacement of RC structures and improves their repairability [16,17].

However, a defect of FRP composites is that their compressive behavior is usually considered to be lower than their tensile strength [18]. This limits their use in concrete columns and walls according to current design standards or guidelines [19,20]. For SFCBs, their compressive strength varies with a number of factors, such as the slenderness ratio (or spiral spacing in concrete members), post-yield stiffness ratio, loading method (monotonic or cyclic loading), and constraint method (non-constrained as a single rod or constrained in concrete members). Generally, a compressive strength of 50% of their tensile strength is commonly considered for SFCBs [21,22,23]. However, this conclusion was drawn from tests on SFCBs under non-confinement or low-stiffness-confinement (with low elastic modulus) conditions. High-stiffness-confinement (with high elastic modulus, such as with a carbon fiber-reinforced polymer (CFRP) composite or steel confinement) can effectively restrain the lateral deformation of the longitudinal reinforcement and improve its compressive strength in concrete members [23,24,25]. This effect on the compressive strength of SFCBs has never been studied before. In addition to the compressive strength of SFCBs in the material phase, their design method in SFCB-reinforced concrete structures is also crucial for achieving a safe design. However, because existing design equations for FRP-RC columns were generally developed assuming FRP is an elastic material, their applicability to SFCB-reinforced concrete columns needs to be verified.

This paper presents an experimental study on the compressive performance of longitudinal SFCBs in concrete cylinders confined with different type of FRP composites. Three types of concrete cylinders reinforced with (or without) longitudinal SFCBs and different types of transverse FRP confinements were designed and tested under monotonic compression. The failure process of the test cylinders was discussed, as well as their stress–strain curves. The compressive strength and possible failure strain of SFCBs were thoroughly analyzed and discussed. Three types of existing design equations (with different approaching methods to evaluate the load-carrying capacity of FRP-reinforced concrete cylinders) were evaluated with the test results in this paper.

## 2. Experimental Program

### 2.1. Specimen Design

The experiment investigated the effect of different FRP composites (used as confinements) on the compressive performance of longitudinal steel-fiber-reinforced concrete (SFCB) cylinders. Three types of concrete cylinders were designed: (1) plain concrete cylinders (C); (2) concrete cylinders reinforced with longitudinal SFCBs and transverse FRP composites, i.e., SFCB and CFRP sheet-reinforced cylinders (SF-CFs) and SFCB and CFRP strip-reinforced cylinders (SF-CDs); and (3) concrete cylinders confined with only transverse FRP composite, i.e., CFRP sheet-confined cylinders (CFs) and CFRP strip-confined cylinder (CDs). The CF and CD cylinders were used as control cylinders for the SF-CF and SF-CD cylinders, respectively. The concrete cylinder was 300 mm high and had a cross-sectional diameter of 150 mm. The thickness of the concrete cover was 15 mm. For comparison, two previously tested concrete cylinders [23] were also included: one was reinforced with longitudinal SFCB and transverse BFRP spiral (ch-f-2); the other one was only confined by a transverse BFRP spiral (f-2). These cylinders are referred to as SF-BS and BS, respectively, in Table 1. The cross-sections of each type of cylinder are illustrated in Figure 3.

The design details of all cylinders are shown in Table 1, where *ρ_l_* is the longitudinal reinforcement ratio; *E_l_* is the transverse confinement stiffness as in Equation (1), in which *E_f_* is the elastic modulus of FRP composites; *ρ_f_* is the transverse reinforcement ratio as in Equation (2), in which *t_f_* is the thickness of the CFRP sheet/strip (=0.167 mm); *b_f_* is the widths of the CFRP sheet (=300 mm) and CFRP strip (=36 mm); *D* is the diameter of the concrete column (=150 mm); *s_f_* is the spacing of the CFRP strip (=18 mm); *A_ss1_* is the cross-sectional area of the BFRP spiral (=50.2 mm^2^); *d_cor_* is the diameter of the concrete core of the cylinder (=120 mm); *s* is the spacing of the BFRP spiral (=23 mm); and *f_l_* is the transverse confinement strength as in Equation (3), in which *f_f_* is the tensile strength of FRP composites. The negative sign in Equation (3) indicates that it is in a state of passive constraint, which is opposite of the direction of lateral expansion.
(1)$$E_{l}=\frac{\rho_{f}E_{f}}{2}$$
(2)$$\rho_{f}=\left\{\begin{array}{c} \frac{4t_{f}}{D}\cdot \frac{b_{f}}{b_{f}+s_{f}}\\ \frac{4A_{ss1}}{d_{cor}s} \end{array}\right.\begin{array}{c} \left(For\;sheet\;or\;strip\right)\\ \left(For\;spiral\right) \end{array}$$
(3)$$f_{l}=-\frac{\rho_{f}f_{f}}{2}$$

For the SF-CF, SF-CD, and SF-BS cylinders, four longitudinal SFCBs were embedded in each cylinder, as shown in Figure 1b,c. The transverse confinement stiffness of the FRP confinement is a crucial factor for the confined concrete strength [26]. This could potentially affect the compressive performance of the SFCBs. Therefore, a similar transverse confinement stiffness (from 1935 to 2155, as shown in Table 1) was designed for all confined concrete cylinders. The transverse reinforcement settings of SF-CF/CF and SF-CD/CD are shown in Figure 1d and Figure 1e, respectively.

### 2.2. Material Properties

#### 2.2.1. SFCB

The SFCB in this paper is pultruded with an inner steel bar (diameter = 10 mm) and an outer FRP layer composed of 85 bundles of 2400-tex basalt fiber and epoxy resin. The mass ratio of the fibers was approximately 85% of the outer FRP layer. The mechanical properties of the SFCB and its components are shown in Table 2, where *D* refers to the diameter; *D_eq_* refers to the equivalent diameter of the SFCB, which is calculated based on the principle of equivalent flexural stiffness [22]. The characteristic strength and modulus values of the SFCBs were obtained from the test results according to ASTM D7205M-06 [27]. Note that the equivalent diameter of the SFCB was specifically used for calculating its characteristic strength and modulus values. The tensile and compressive behavior of the SFCB is shown in Figure 2.

#### 2.2.2. CFRP Sheet/CFRP Strip/BFRP Spiral

The physical and mechanical properties of the CFRP sheet, CFRP strip, and BFRP spiral are presented in Table 3. The CFRP sheet was manufactured by TORAY INDUSTRIES, INC. Tokyo, Japan (product No. UT70-30G) [28]. The CFRP strip was hand cut from the CFRP sheet, and therefore its physical and mechanical properties were identical to those of the CFRP sheet. The characteristic strength and modulus values of the BFRP spiral were minimum values guaranteed by the manufacturer.

#### 2.2.3. Concrete

The mechanical properties of the concrete were obtained through tests according to ASTM C469-14 [29] and ASTM C39M-21 [30]. The elastic modulus and compressive strength of the concrete (c) were 29 GPa and 43 MPa, respectively.

### 2.3. Test Instruments and Setup

The compressive loading test was conducted on a servo compression machine with a loading capacity of 3000 kN, as shown in Figure 4a. A thin gypsum cushion was placed on both the top and bottom surfaces of the specimen to ensure a uniform distribution of the compressive load. The loading process was displacement-controlled with a loading speed of 0.006 mm/s.

Four linear variable differential transformers (LVDTs) were used to capture the compressive displacement, as shown in Figure 4. The compressive strain was calculated as the mean value of the compressive displacements divided by the height of the column. The compressive loading stress was calculated as the loading force (obtained from the servo compression machine) divided by the cross-section area of the column.

Four axial strain gauges (gauge length = 2 mm) were used to capture the compressive strains of the SFCBs. The two strain gauges on each SFCB were set on its inner and outer surfaces, as shown in Figure 4b,c, and the mean value of these two strains was regarded as the compressive strain of the SFCB before bending. Two transverse strain gauges (gauge length = 2 mm) were used to capture the compressive strains of the BFRP spirals, as shown in Figure 4c. Four strain gauges (gauge length = 20 mm) were placed at 90° around the circumference of the column in both the axial and hoop directions to capture the initial axial and hoop strain of the column, as shown in Figure 4b,c. All the strain gauges were placed at the mid-height of the column.

## 3. Test Results and Discussion

### 3.1. General Observation and Stress–Strain Curve of Cylinders

The stress–strain curves and the typical failure modes of the cylinders are shown in Figure 5 and Figure 6, respectively.

#### 3.1.1. For SF-CF and CF Cylinders

The stress–strain curve of the SF-CF cylinder increased linearly during the initial loading, until a turning point occurred at a strain of 0.0043. After the turning point, the stress continued to increase almost linearly but with a relatively small slope with the development of the strain, as shown in Figure 5a. This indicates that a strain-hardening characteristic was achieved for the SF-CF cylinder. No obvious damage was visualized for the SF-CF cylinder before the loading capacity of the servo compression machine was reached, as shown in Figure 6a.

The stress–strain curve of the CF cylinder was similar to that of the SF-CF cylinder. However, after the turning point, the compressive stress of the CF cylinder increased less stably than that of the SF-CF cylinder, but with a fluctuatingly increasing pattern, as shown in Figure 5a. This was potentially due to the local crushing of the confined concrete, which led to stress redistribution in the cylinder, as shown in Figure 6b. The CFRP sheet remained intact at the same time. This indirectly proved that a relatively low confinement effect was achieved by the CF cylinder compared to the SF-CF cylinder, which achieved a relatively strong confinement effect owing to the combination of the SFCBs and CFRP sheet.

#### 3.1.2. For SF-CD and CD Cylinders

The stress–strain curve of the SF-CD cylinder increased linearly during the initial loading stage until a turning point was reached, as shown in Figure 5b. At a strain of 0.0149, a few microcracks appeared on the concrete section, between the CFRP strips. As the loading process continued, the stress–strain curve continued to increase with a relatively small slope (compared to the slope in the initial loading stage), and the concrete cracks continued to increase and dilate. At a strain of around 0.042, a small cracking sound was captured, and the stress–strain curve showed a short drop. This was possibly due to the fracture or splitting of the outer FRP layer of the SFCB. At a strain of around 0.048, concrete crushing was visible between the CFRP strips, as shown in Figure 6c. The stress–strain curve then dropped sharply, indicating a total failure of the cylinder.

The failure process of the CD cylinder was similar to that of the SF-CD cylinder. The CD cylinder also failed in a concrete crushing mode, with the CFRP strips remaining intact after the test. This can be seen in Figure 6d. The compressive stress of the CD cylinder was higher than that of the SF-CD cylinder at the same compressive strain, as shown in Figure 5b. This may be due to the fact that the longitudinal reinforcements have an adverse impact on the compressive strength of the confined concrete section. The longitudinal reinforcements can weaken the integrity of the section and potentially reduce its effectively confined area. This effect could be more pronounced for the SF-CD than for the SF-CF, since the effectively confined area of the SF-CD cylinder is smaller than that of the SF-CF cylinder due to the distributed confinement mode of the SF-CD. It could exacerbate the unfavorable impact of the longitudinal reinforcements on the compressive strength of the confined concrete section.

#### 3.1.3. For the SF-BS and BS Cylinders

The stress–strain curves of the SF-BS and BS cylinders have been reported in previous studies [23]. These cylinders were confined with inner BFRP spirals, which resulted in different failure processes from the outer-confined cylinders, such as the SF-CF, SF-CD, etc. A few vertical cracks appeared when the strain reached around 0.004, and then a turning point was observed on the stress–strain curve, as shown in Figure 5c. After the turning point, the stress–strain curve continued to increase with a relatively small slope (compared to the slope at the beginning). The vertical cracks expanded to the top and bottom of the outer surface of the cylinder. When the strain reached around 0.015, a few continuous fracture sounds were captured. This was presumably due to the BFRP spiral fracture, as shown in Figure 6e. After that, the vertical cracks expanded rapidly with the development of the loading process, and the concrete cover spalled gradually. For the SF-BS cylinder, a fracture sound was further captured when the strain reached around 0.016. This was possibly due to the fracture of the SFCB, as shown in Figure 6e.

The characteristic values of the stress–strain curves of the cylinders are shown in Table 1. Note that the peak stresses of the SF-CF and CF cylinders were indicated with the maximum loading stresses of the servo loading machine (=3000 kN), instead of the actual peak stresses when those cylinders failed. A strain-hardening characteristic [23] was revealed for all confined cylinders due to their well-confined conditions. The peak stresses and strains of all confined cylinders were significantly higher than those of the unconfined (plain) concrete cylinder. Especially for the cylinders confined with outer CFRP sheet/strips (i.e., SF-CF, SF-CD, CF, CD), their peak stresses and strains reached approximately three times the corresponding values of the inner BFRP spiral-confined concrete cylinders (i.e., SF-BS, BS) due to their enlarged effectively confined areas and the high-efficiency confinement method. This superior peak strain of the outer confined cylinder could be useful to maximize the compressive performance of the embedded SFCBs.

### 3.2. Compressive Stress–Strain Curves of SFCBs in Concrete Cylinders Confined with Different FRP Composites

#### 3.2.1. Calculation Method of Compressive Stress–Strain Curve of SFCB

The compressive stress–strain relationship of the embedded SFCB was obtained by subtracting the compressive stress of the FRP composites-confined concrete (for example, CF) cylinder from the compressive stress of the longitudinal SFCBs and transverse FRP composites-reinforced (for example: SF-CF) cylinder, as shown in Figure 5c. The calculation method is provided in Equation (4), where *A_col_* is the cross-sectional area of the concrete column, and *A_sfeq_* is the equivalent cross-sectional area of the SFCB. Since the recorded compressive strain step could vary for different cylinders, a normalization was first conducted in MATLAB for each cylinder to obtain the same compressive strain step of 0.001. The calculated stress–strain curves of the SFCB are shown in Figure 7, in which the tensile stress–strain curve of a SFCB (tested as a single rod) is also provided for comparison. It is worth noting that the stress–strain curve of the SFCB in SF-CD was not included in Figure 7 because the stress of the CD cylinder was generally higher than that of the SF-CD cylinder in each strain step, and as a result, a positive stress–strain curve could not be extracted by applying Equation (4).

The compressive stress–strain curve of a SFCB in SF-BS cylinders fits relatively well with the corresponding tensile curve, especially at the initial loading stage and at the turning point, which indicates the yield of the inner steel bar of the SFCB. This shows the actual compressive behavior of SFCBs in concrete cylinders, which has been explained in previous studies [23]. The initial stress–strain curve of SFCBs in SF-CF cylinders was also generally consistent with the trend of the corresponding tensile curve. However, there is a large deviation between the turning points of these two curves, as shown in Figure 7. This could be attributed to the following two reasons: (1) The difference in compressive stress between the confined concrete part in SF-CF and CF cylinders at the same compressive strain state can be attributed to the different confinement effects of the longitudinal and transverse confinement in SF-CF and the single transverse confinement in CF. In SF-CF cylinders, the longitudinal confinement provides additional lateral support to the concrete, which increases the compressive strength of the confined concrete part. This is not the case in CF cylinders, where the concrete is only confined in the transverse direction. As a result, the compressive stress of the confined concrete part in SF-CF cylinders could be slightly higher than that in CF cylinders at the same compressive strain state. (2) On the other hand, the presence of SFCBs in a SF-CF cylinder could disrupt the integrity of the confined concrete section, which could diminish the compressive performance of the confined concrete. This phenomenon is fundamentally inconsistent with the findings of previous studies, which demonstrated that the initial compressive behavior of a SFCB should be consistent with its tensile performance, both under unconfined compression and in concrete cylinders [22]. As a result, the calculated compressive stress–strain curve of SFCBs in SF-CF cylinders cannot accurately represent the actual compressive behavior of a SFCB, and the curve must be modified.
(4)$$\sigma_{sf}^{-}=\frac{(\sigma_{sf-cf/sf-bs}-\sigma_{cf/bs})A_{col}}{4A_{sfeq}}$$

#### 3.2.2. Modification for the Calculated Compressive Stress–Strain Curve of SFCBs

The unique post-yield stiffness behavior of a SFCB ensures that a turning point will occur on its stress–strain curve under both tensile and compressive loading once the loading strain exceeds the yield strain of its inner steel bar. It is also well known that the tensile and compressive behavior of steel bars are almost identical when well confined in concrete structures. As a result, the turning point of the compressive stress–strain curve of a SFCB should coincide with its tensile stress–strain curve. Based on this principle, a modification was made to the calculated stress–strain curve of the SFCB by adding a fixed value to both the stress and strain in order to “move” it along the tensile stress–strain curve until the turning points of the compressive and tensile stress–strain curves coincided, as shown in Figure 8.

#### 3.2.3. Compressive Mechanism of SFCBs in Different FRP Composites-Confined Concrete Cylinders

The stress–strain curves of SFCBs in different types of FRP composites-confined concrete cylinders are shown in Figure 9. The modified stress–strain curve generally reflects the actual compressive behavior of the SFCB, particularly at the initial loading stage, where it fits well with the corresponding tensile curve.

After the turning point, each compressive stress–strain curve of the SFCB did not follow the corresponding tensile curve but developed with a relatively low post-yield stiffness. This agrees well with previous findings [23]. However, the post-yield stiffness of the confined SFCB developed relatively higher when it was confined with a relatively high elastic modulus of FRP composite (CFRP sheet in SF-CF cylinder) than when it was confined with a relatively low elastic modulus of FRP composite (BFRP spiral in SF-BS cylinder). This is because the high elastic modulus of FRP confinement can effectively restrict the transverse displacement of the SFCB in concrete cylinders, and hence improve its compressive performance after yield.

In addition, the CFRP sheet in the SF-CF cylinder provided superior confinement, resulting in a significantly longer compressive stress–strain curve for the SFCB than in the SF-BS cylinder. The possible SFCB failure strain in SF-CF reached almost 0.03, which is around 88% of the tensile peak strain (=0.034). Although the actual compressive peak stress of the SFCB in SF-CF could not be directly obtained from the modified stress–strain curve, the prolonged stress–strain curve (or increased compressive peak strain) suggests that the SFCB was able to utilize a relatively high percentage of its material strength when confined with relatively high elastic modulus FRP composites.

### 3.3. Compressive Peak Stress of SFCBs Embedded in Concrete Cylinders Confined with Different FRP Composites

The compressive peak stress of the SFCB (*σ_p_sf_*
^−^) can be calculated by adding the maximum stress obtained from the stress–strain curve of the SFCB [i.e., the maximum stress value of SFCB (${\left|\sigma_{sf}{}^{-}\right|}_{\max}$) in Figure 9 to the complementary compressive contribution of the concrete (with an area equal to the cross-sectional area of the SFCBs in the cylinder). The calculation method is provided in Equation (5) [23], where *A_sf_* is the cross-sectional area of the SFCB.
(5)$$\sigma_{p_{-}sf}{}^{-}={\left|\sigma_{sf}{}^{-}\right|}_{\max}+\frac{\sigma_{c}A_{sf}}{A_{sfeq}}$$

A comparison of the compressive peak stress ratio [i.e., ratio of compressive peak stress to tensile strength (*σ_p_sf_*
^+^)] of SFCBs tested in different conditions is shown in Figure 10.

It has been shown in previous studies that decreasing the spacing of the spiral can help increase the compressive strength of SFCBs in concrete cylinders [23]. As shown in Figure 10, the compressive peak stress ratio increases with the decrease of the spiral spacing.

The same conclusion can be drawn from Figure 10 by comparing the compressive peak stress ratio of the SFCB in the SF-CF cylinder to the compressive peak stress ratios of SFCBs in spiral-confined cylinders. The compressive peak stress ratio of the SFCB in the SF-CF cylinder (fully covered without spacing) reached 0.43, which is 16% higher than the mean value of the compressive peak stress ratios of SFCBs in concrete cylinders confined by inner spirals with different spacings (0.37). However, it also indicates that this improvement is not significant by simply decreasing the spacings of the confinements. The maximum compressive peak stress of SFCBs can still be approximately defined as 50% of their tensile strength, as in previous findings [22,23,31].

It is worth noting that the failure strain of the SFCB in the SF-CF cylinder reached almost 0.03, which indicates that a potentially higher compressive peak stress (compared to the calculated compressive peak stress in this section) could be achieved in the test. However, due to the lack of sufficient experimental data, the actual compressive peak stress of the SFCB confined in high-stiffness FRP composites, as well as the factors affecting it, remain unknown.

### 3.4. Evaluation of Design Equations for Load-Carrying Capacity of SFCB-Reinforced Concrete Cylinders

The design method for the load-carrying capacity of SFCB-reinforced concrete cylinders is crucial to realizing a precise and cost-effective design of SFCB-reinforced concrete structures. However, it has not been studied before. Since the elastic failure mode (determined by the outer fiber layer failure) of a SFCB is similar to that of FRP reinforcement in concrete cylinders, the design equations for predicting the load-carrying capacity of FRP-reinforced concrete cylinders can be used to evaluate the load-carrying capacity of SFCB-reinforced concrete cylinders.

There are generally three types of design equations for evaluating the load-carrying capacity of FRP-reinforced concrete cylinders, considering the variable compressive contributions of the longitudinal reinforcements for the concrete cylinders:

Method I: Considering the contribution of longitudinal reinforcement with a strength-reduction factor.

Tobbi et al. [31] set the compressive strength of FRP bars to 35% of their tensile strength through an experimental study of GFRP bars-reinforced concrete columns. They obtained Equation (6), where *P_pre_* is the predicted load-carrying capacity of the reinforced concrete cylinder; *α*_1_ is the capacity reduction coefficient of the column; *f′_c_* is the compressive strength of the confined concrete; *A_g_* is the cross-sectional area of the column; and *α_sf_* is the strength reduction coefficient of the longitudinal reinforcement.
(6)$$P_{pre}=\alpha_{1}{f^{’}}_{c}(A_{g}-A_{sf})+\alpha_{sf}{\sigma_{p_{-}sf}}^{+}A_{sfeq}\;\mathrm{where}\;\alpha_{1}=0.85\;\mathrm{and}\;\alpha_{sf}=0.35$$

Afifi et al. [32] conducted an experimental study on the axial bearing capacity of concrete columns longitudinally reinforced with CFRP bars. They proposed a strength reduction factor of 0.25 for predicting the compressive contribution of the longitudinal reinforcement, as in Equation (7).
(7)$$P_{pre}=\alpha_{1}{f^{’}}_{c}(A_{g}-A_{sf})+\alpha_{sf}{\sigma_{p_{-}sf}}^{+}A_{sfeq}\;\mathrm{where}\;\alpha_{1}=0.85\;\mathrm{and}\;\alpha_{sf}=0.25$$

Method II: Considering the contribution of longitudinal reinforcement with a maximum compressive strain.

Mohamed et al. [33], Hadi et al. [34], Hadhood et al. [35], and Xue et al. [36] conducted experimental studies on the compressive behavior of GFRP bar-reinforced concrete cylinders. They proposed the maximum compressive strains of FRP bars to be 0.002, 0.003, 0.0024, and 0.002, respectively, as shown in Equations (8)–(11).
(8)$$P_{pre}=\alpha_{1}{f^{’}}_{c}(A_{g}-A_{sf})+0.002E_{f}A_{sfeq}\;\mathrm{where}\;\alpha_{1}=0.85$$
(9)$$P_{pre}=\alpha_{1}{f^{’}}_{c}(A_{g}-A_{sf})+0.003E_{f}A_{sfeq}\;\mathrm{where}\;\alpha_{1}=0.85$$
(10)$$P_{pre}=\alpha_{1}{f^{’}}_{c}(A_{g}-A_{sf})+0.0024E_{f}A_{sfeq}\;\mathrm{where}\;\alpha_{1}=0.85$$
(11)$$P_{pre}=\alpha_{1}{f^{’}}_{c}A_{g}+0.002E_{f}A_{sfeq}\;\mathrm{where}\;\alpha_{1}=0.85$$

Method III: Considering the contribution of longitudinal reinforcement with its actual compressive strength in concrete cylinders obtained through tests.

Tang et al. [37] proposed a different approach to predict the load-carrying capacity of FRP-reinforced concrete columns by considering the actual compressive strength of the FRP bars in concrete cylinders obtained from tests. The design equation is provided in Equation (12), where *f′_cc_* is the compressive strength of concrete with only transverse reinforcement.
(12)$$P_{pre}=\beta {f^{’}}_{cc}(A_{g}-A_{sf})+{\sigma_{p_{-}sf}}^{-}A_{sfeq}\;\mathrm{where}\;\beta =0.85$$

Note that the value of *f′_c_* used in this paper is not the compressive strength of plain concrete, but the compressive stress of the CF/CD/BS cylinders that corresponds to the failure strain of SFCBs in the SF-CF/SF-CD/SF-BS cylinders, respectively. This is done to offset the confinement concrete effect. The predicted-to-experimental ratios of different design equations are shown in Figure 11 and Table 4, where *P_exp_* is the load-carrying capacity of reinforced concrete cylinders from experiments.

The predicted-to-experimental ratios of Methods I and II (0.80~0.96) are generally lower than 1, which indicates that Methods I and II underestimated the load-carrying capacity of SFCB-reinforced concrete cylinders. This is potentially due to the fact that the combination of longitudinal SFCB and transverse FRP composite restraint exerts stronger confinement on concrete in SF-CF/SF-CD/SF-BS cylinders than in the case of only transverse FRP composite restraint in CF/CD/BS cylinders. The actual compressive contribution of concrete for SF-CF/SF-CD/SF-BS cylinders may be larger than that of the corresponding CF/CD/BS cylinders. Therefore, the compressive contribution of concrete in SF-CF/SF-CD/SF-BS cylinders may be underestimated when the compressive strength of concrete in CF/CD/BS cylinders is predicted. This conclusion can be further supported by comparing the predicted-to-experimental ratios of SF-CF cylinders to those of SF-CD cylinders. The predicted-to-experimental ratios of SF-CF cylinders vary from 0.80 to 0.88, which are generally lower than those of SF-CD cylinders, which vary from 0.89 to 0.94. This is because the relatively strong confinement effect for the concrete in SF-CF cylinders generates more errors in the estimation of the compressive contribution of concrete than those in SF-CD cylinders.

The average predicted-to-experimental ratio of the two equations in Method I is 0.90, which is higher than the average predicted-to-experimental ratio of 0.84 for the four equations in Method II. This finding is consistent with the results of previous studies [23] on FRP-reinforced concrete cylinders, which showed that the strength reduction factor is a more accurate way to predict the load-carrying capacity of FRP-reinforced concrete cylinders than the ultimate compressive strain of concrete. This may be due to the fact that the prediction of the ultimate compressive strain of concrete is more suitable for the condition that the concrete cylinder is damaged due to the compression of the concrete. However, this condition is not applicable to the concrete cylinders damaged due to the failure of FRP bars (or SFCBs).

Method III provides the most accurate prediction of the load-carrying capacity of SFCB-reinforced concrete cylinders, with an average predicted-to-experimental ratio of 0.96. This is likely due to the fact that Method III considers the actual compressive strength of SFCBs in concrete cylinders. However, this method requires the compressive strength of SFCBs to be obtained in advance. Therefore, further studies are required on the compressive behavior of different SFCBs embedded in concrete cylinders to improve its applicability and accuracy.

## 4. Conclusions

The compressive performance of SFCBs confined by different type of FRP composites in concrete cylinders were experimentally studied in this paper. The following conclusions can be drawn:The post-yield stiffness of the confined SFCBs developed to become relatively higher when it was confined with a relatively high elastic modulus CFRP composite than when it was confined with a relatively low elastic modulus of BFRP composite.The compressive failure strain of the SFCB in the SF-CF cylinder could have reached 88% of its tensile peak strain, which indicates that a relatively high utilization of the material strength may be achieved for the SFCB restrained with relatively high elastic modulus confinements.The design equations that consider the compressive contribution of SFCBs in concrete cylinders with a strength reduction factor of the SFCB (in Method I) or a maximum compressive strain of concrete (in Method II) generally underestimated the load-carrying capacity of SFCB-reinforced concrete cylinders. In addition, this underestimation appears more significant for the cylinder with a relatively strong confinement.The approaching method, which applies an actual compressive strength of the SFCB for considering its compressive contribution in concrete cylinders (in Method III), gave the most accurate prediction for the load-carrying capacity of SFCB-reinforced concrete cylinders. However, its applicability and accuracy need to be verified with more experimental data in the future.

## Figures and Tables

**Figure 1 polymers-15-04051-f001:**
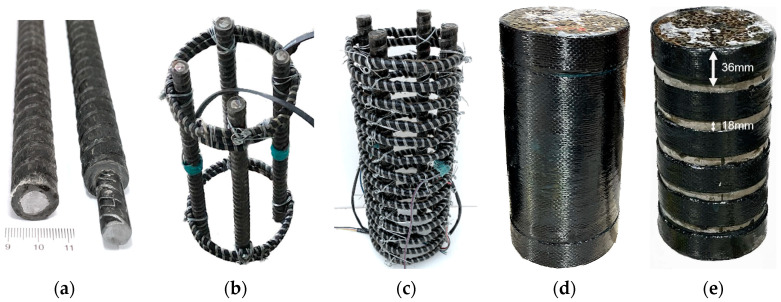
SFCB and reinforcement details of different cylinders: (**a**) SFCB; (**b**) Longi. reinf. of SF-CF/SF-CD cylinders; (**c**) Longi. and trans. reinf. of SF-BS cylinder; (**d**) Trans. reinf. of SF-CF/CF cylinders; (**e**) Trans. reinf. of SF-CD/CD cylinders.

**Figure 2 polymers-15-04051-f002:**
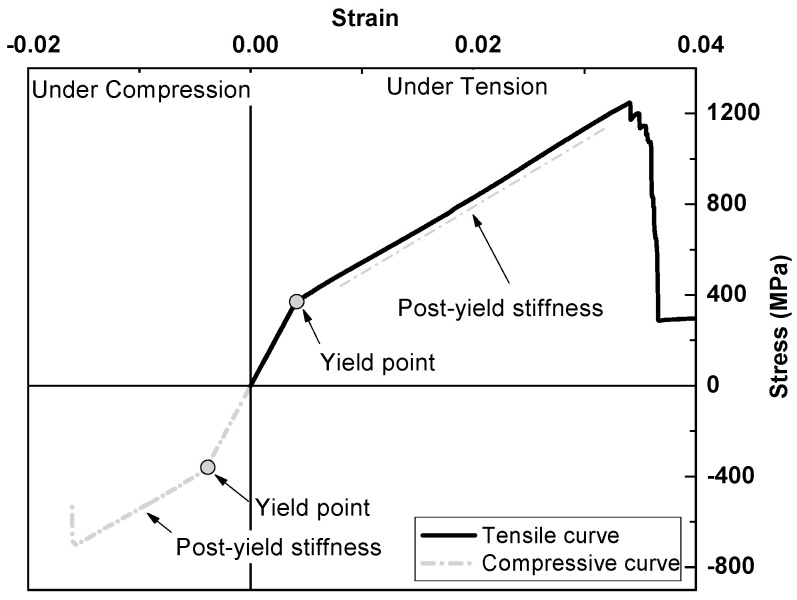
Tensile and compressive stress–strain curves of the SFCB tested as a single rod under monotonic loading.

**Figure 3 polymers-15-04051-f003:**
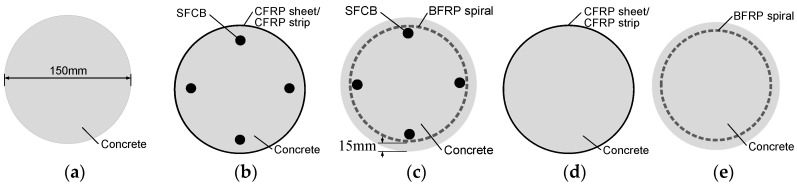
Cross-sections of test cylinders: (**a**) C; (**b**) SF-CF/SF-CD; (**c**) SF-BS; (**d**) CF/CD; (**e**) BS.

**Figure 4 polymers-15-04051-f004:**
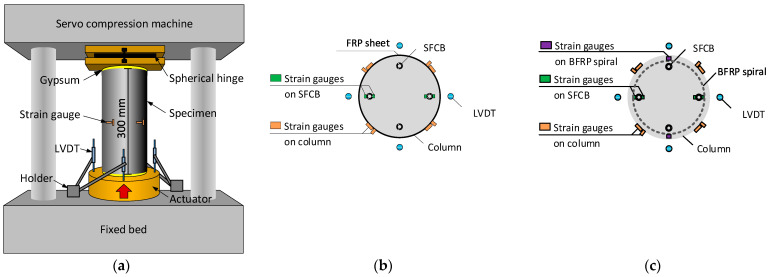
Test setup: (**a**) Test instruments; (**b**) Strain gauge distribution in SF-CF/SF-CD cylinder; (**c**) Strain gauge distribution in SF-BS cylinder.

**Figure 5 polymers-15-04051-f005:**
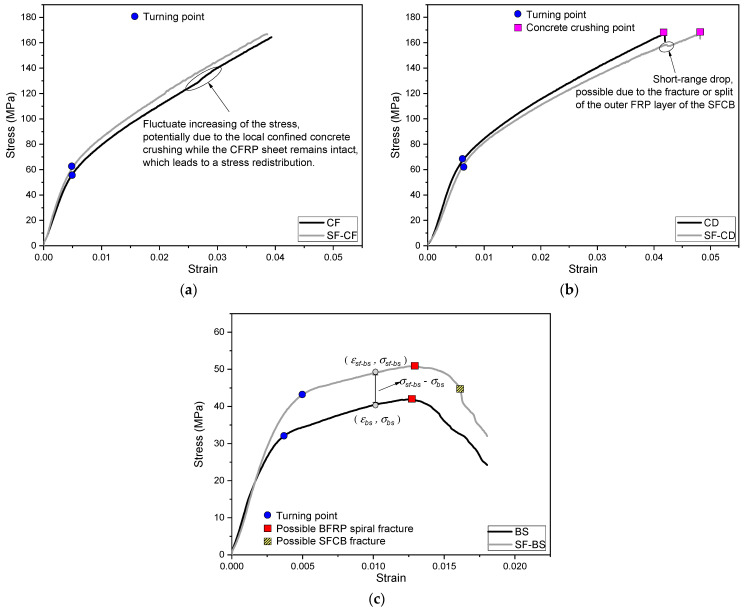
Stress–strain curves of different types of columns: (**a**) CF type/SF-CF type; (**b**) CD type/SF-CD type; (**c**) BS type/SF-BS type [23].

**Figure 6 polymers-15-04051-f006:**
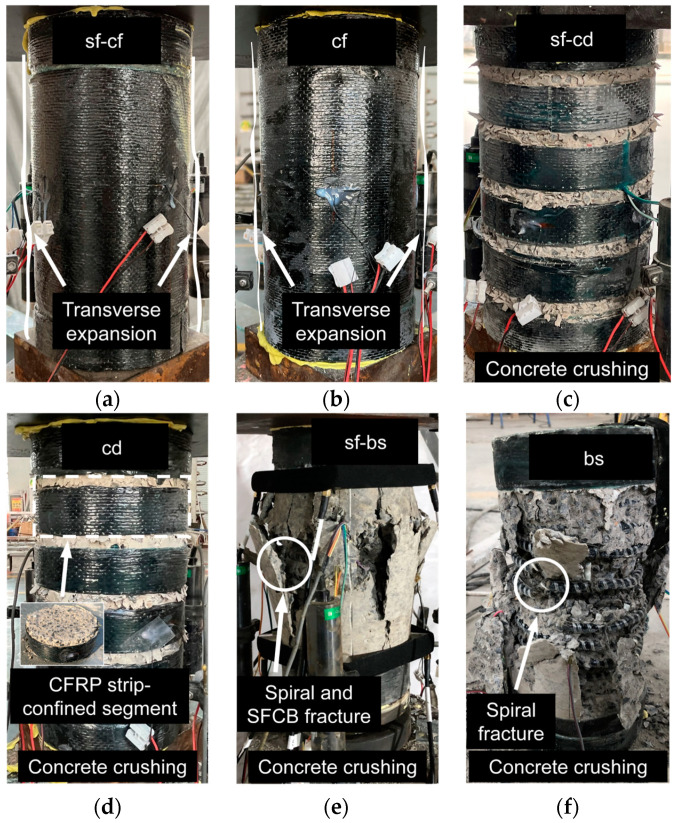
Failure modes of cylinders: (**a**) SF-CF; (**b**) CF; (**c**) SF-CD; (**d**) CD; (**e**) SF-BS; (**f**) BS.

**Figure 7 polymers-15-04051-f007:**
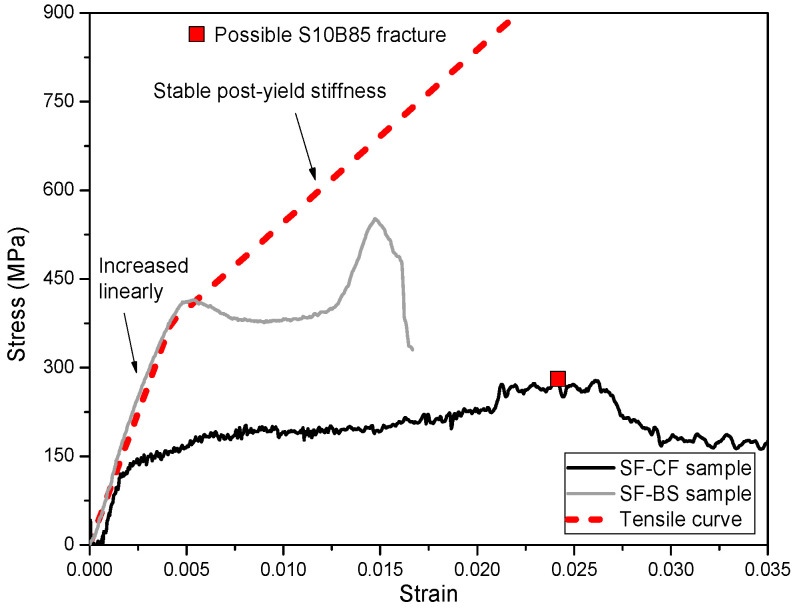
Stress–strain curves of SFCBs in different cylinders.

**Figure 8 polymers-15-04051-f008:**
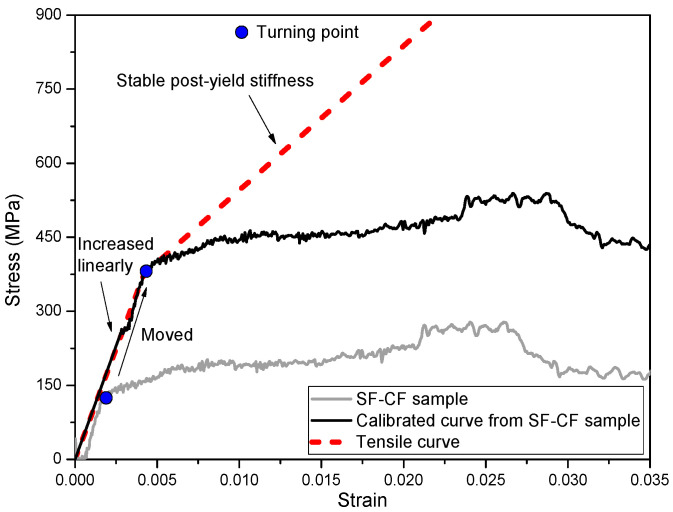
Modification method for the calculated stress–strain curve of SFCB in SF-CF cylinder.

**Figure 9 polymers-15-04051-f009:**
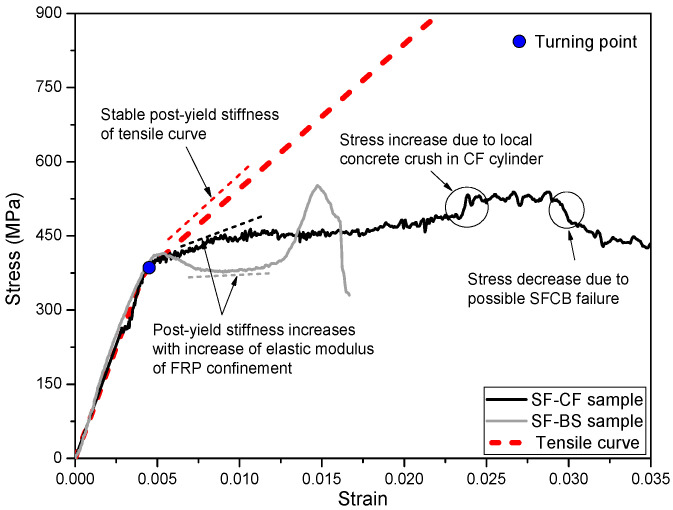
Stress–strain curves of SFCBs in different cylinders after modification.

**Figure 10 polymers-15-04051-f010:**
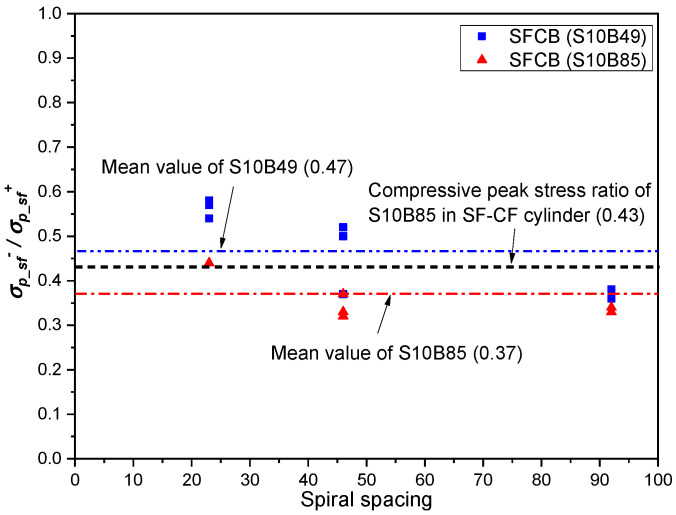
Comparison of the compressive peak stress ratio of SFCBs in concrete cylinders confined by spirals with different spacings.

**Figure 11 polymers-15-04051-f011:**
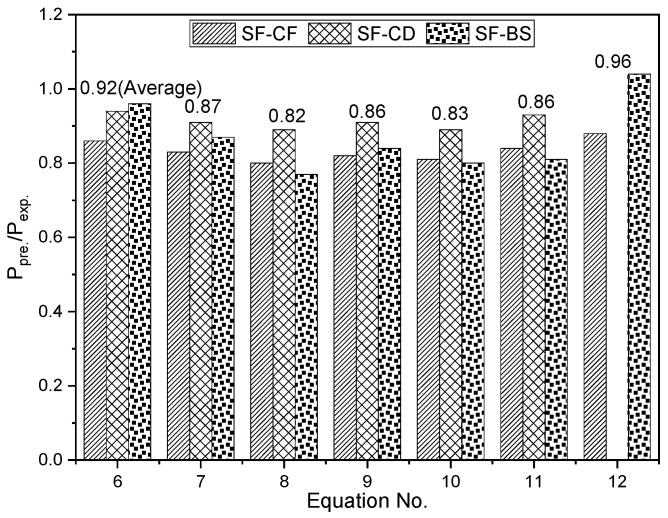
Predicted-to-experimental ratios of the design equations for predicting the test cylinders.

**Table 1 polymers-15-04051-t001:** Design details and test results of cylinders.

Cylinder Type	Longi. Reinf.		Trans. Reinf.							Cylinder		
	Type	*ρ_l_* (%)	Type	Layer	*b_f_* or *d_f_* (mm)	*s_f_* or *s* (mm)	*E_l_* (MPa)	*ρ_f_* (%)	*f_l_* (MPa)	Failure Mode	Peak Stress (MPa)	Peak Strain
C	—	—	—	—	—	—	—	—	—	CC	43	0.004
SF-CF	SFCB	3.0	cs	4	300	—	2155	1.8	36.5	—	170	0.039
SF-CD	SFCB	3.0	cd	6	36	18	2155	1.8	36.5	SFF-CC	167	0.048
SF-BS	SFCB	3.0	bs	—	8	23	1935	7.3	46.8	SF-SFF	54	0.015
CF	—	—	cs	4	300	—	2155	1.8	36.5	—	170	0.039
CD	—	—	cd	6	36	18	2155	1.8	36.5	CC	167	0.042
BS	—	—	bs	—	8	23	1935	7.3	46.8	SF	43	0.013

Note: cs = CFRP sheet; cd = CFRP strip with a certain distance; bs = BFRP spiral; CC = concrete crush; SFF = SFCB split or fracture; SF = BFRP spiral fracture. The data of SF-BS and BS cylinders are from the ch-f-2 and f-2 specimens, respectively, from a previous paper [23].

**Table 2 polymers-15-04051-t002:** Mechanical properties of longitudinal reinforcement and its components.

Type	*D* (mm)	*D_eq_* (mm)	Elongation (%)	Density (g/cm^3^)	Yield Strength (MPa)	Tensile Strength (MPa)	Elastic Modulus (GPa)	Post-Yield Modulus (GPa)
SFCB	16.8	12.9	4.5	—	376	1247	92	29
Inner steel bar	10.0	—	14.3	7.85	400	528	200	—
Basalt fiber	0.013	—	2.5	2.63	—	2250	90	—
Epoxy resin	—	—	6.1	1.06	—	95	3.6	—

**Table 3 polymers-15-04051-t003:** Physical and mechanical properties of transverse reinforcements.

Type	Fiber Weight (g/m^2^)	Thickness (mm)	Density (g/cm^3^)	Tensile Strength (MPa)	Elastic Modulus (GPa)
CFRP sheet	300	0.167	1.80	4100	242
CFRP strip	300	0.167	1.80	4100	242
BFRP spiral	—	—	2.00	1281	53

**Table 4 polymers-15-04051-t004:** Predicted-to-experimental ratio results of the design equations for predicting the test cylinders.

Equation No.	Approaching Method	$P_{pre.}/P_{\exp.}$				
		SF-CF	SF-CD	SF-BS	Average	Deviation
6	I	0.86	0.94	0.96	0.92	0.04
7	I	0.83	0.91	0.87	0.87	0.03
8	II	0.80	0.89	0.77	0.82	0.05
9	II	0.82	0.91	0.84	0.86	0.04
10	II	0.81	0.89	0.80	0.83	0.04
11	II	0.84	0.93	0.81	0.86	0.05
12	III	0.88	—	1.04	0.96	0.08

## Data Availability

The data presented in this study are available on request from the corresponding author.

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
