# Peer review of "Compressive Performance of Longitudinal Steel-FRP Composite Bars in Concrete Cylinders Confined by Different Type of FRP Composites"

_polymers, 2023, doi:10.3390/polym15204051_

Round 1

Reviewer 1 Report

In this paper, the respected authors have conducted experimental tests on the compressive performance of longitudinal steel-fiber-reinforced polymer composite bars (SFCBs) in concrete cylinders confined by different type of fiber-reinforced polymer (FRP) composites. The subject of this paper is interesting. This paper can be published pending addressing the following comments and questions.

Comments:

1) Innovation and contribution of this paper is not clear. There are hundreds of papers is literature about the response of concrete structures reinforced with steel-FRP bars that are strengthened with FRP-sheets.

2) Tensile strength of FRP rods is not good and they are usually used as tension reinforcing elements. In this paper, FRP rods are employed in concrete cylinder under compressions. Why?

3) It is not clear that curves depicted in different figures with title "tension curve" represent the structural behavior of which type of cylindrical samples?

4) In legend of figures, please use "SF-CF sample" instead of "In SF-CF".

5) The conclusion part is lengthy and it is not supported by data.

6) The number of self-citation in reference list is relatively high.

Minor editing of English language required

Author Response

1) Innovation and contribution of this paper is not clear. There are hundreds of papers is literature about the response of concrete structures reinforced with steel-FRP bars that are strengthened with FRP-sheets.

Response:

Thank you for indicating this issue. While there is a wealth of literature on concrete structures reinforced with steel-FRP bars strengthened with FRP sheets, most of these studies focus on the structural behavior of FRP-RC members rather than the mechanical behavior of the longitudinal reinforcement itself. This paper specifically addresses the compressive behavior of longitudinal SFCBs in concrete columns, which is a critical but understudied area of research. Two major innovations or contributions were provide detailed as follows:

(i) Existing standards (AASHTO 2011; ACI 2006) commonly underestimate or neglect the compressive contribution of longitudinal FRP bars in FRP-RC structures. However, the elastic-plastic behavior of SFCBs, which is similar to that of steel bars (Sun et al. 2011), means that their actual compressive contribution cannot be neglected. Previous studies have shown that the compressive behavior of SFCBs, including their compressive strength, can vary depending on the confinement setting, especially under low-stiffness-confinement conditions (Tang et al. 2021, 2022).

High-stiffness-confinement (e.g., carbon fiber-reinforced polymer (CFRP) composites or steel confinement, which have high elastic moduli) can effectively restrain the lateral deformation of longitudinal reinforcement and improve its compressive strength in concrete members. This effect on the compressive strength of SFCBs has not been studied before and is presented in this paper as the first innovation and contribution.

(ii) The design of FRP-RC columns for compressive strength has been extensively studied, and many design equations have been developed. However, there is no specific design equation for SFCB-reinforced columns, and the existing design equations for FRP-RC structures were generally developed based on the elastic properties of FRP materials. Therefore, their applicability to SFCB-reinforced concrete columns needs to be verified.

This paper provides a verification analysis of a variety of existing design equations using the test results generated in this study. This is the second innovation and contribution of this paper.

To address this issue, more comments combined with original expressions are provided in Introduction, as in L48-54, L56-58, and as follows:

However, this conclusion was drawn from tests on SFCBs under non-confinement or low-stiffness-confinement (with low elastic modulus) condition. High-stiffness-confinement [with high elastic modulus, such as a carbon fi-ber-reinforced polymer (CFRP) composite or steel confinement] can effectively restrain the lateral deformation of the longitudinal reinforcement and improve its compressive strength in concrete members [23-25]. This effect to the compressive strength of SFCBs has never been studied before.

However, because existing design equations for FRP-RC columns were generally developed assuming FRP is an elastic material, their applicability to SFCB-reinforced concrete columns needs to be verified.

2) Tensile strength of FRP rods is not good and they are usually used as tension reinforcing elements. In this paper, FRP rods are employed in concrete cylinder under compressions. Why?

Response:

Thanks for the comments. It is true that FRP rods are typically used as tensile reinforcement, and their compressive contributions are commonly underestimated or neglected by existing standards (AASHTO 2011; ACI 2006). However, SFCBs are different from traditional FRP materials. Due to the composite effect of the inner steel bar and outer FRP layer, SFCBs exhibit relatively strong compressive performance (up to 50% of their tensile strength with relatively large compressive strain, as illustrated below). This makes them more similar to steel bars than to FRP rods. Therefore, the compressive contribution of SFCBs cannot be neglected when they are used to reinforce concrete structures.

3) It is not clear that curves depicted in different figures with title "tension curve" represent the structural behavior of which type of cylindrical samples?

Response:

Thanks for the comment. The “tensile curve” represent the tensile stress-strain curve of SFCB tested as a signle rod. To make it clarify, the caption of Figure 2 is revised to “Figure 2. Tensile and compressive behavior stress-strain curves of the SFCB tested as a single rod under monotonic loading.”

More expressions have been added into revision, as in L259-261 and as follows:

The calculated stress-strain curves of SFCB are shown in Figure 7, in which the tensile stress-strain curve of SFCB (tested as a single rod) is also provided for comparison.

4) In legend of figures, please use "SF-CF sample" instead of "In SF-CF".

Response:

Thank you for the great suggestions. The legends of figure 7~9 have been revised as follows:

Figure 7. Stress-strain curves of SFCB in different cylinders.

Figure 8. Modification method for the calculated stress-strain curve of SFCB in SF-CF cylinder.

Figure 9. Stress-strain curves of SFCB in different cylinders.

5) The conclusion part is lengthy and it is not supported by data.

Response:

Thanks for the comment. The conclusion has been simplipied and restructured as follows:

• The post-yield stiffness of the confined SFCB developed relatively higher when it was confined with a relatively high elastic modulus CFRP composite than when it was confined with a relatively low elastic modulus of BFRP composite.

• The compressive failure strain of SFCB in SF-CF cylinder could have reached 88% of its tensile peak strain, indicates that a relatively high utilization of the material strength may be achieved for the SFCB restrained with relatively high elastic modulus confinements.

• The design equations that consider the compressive contribution of SFCBs in concrete cylinders with a strength reduction factor of SFCB (in Method I) or a maximum compressive strain of concrete (in Method â…¡) generally underestimated the load-carrying capacity of SFCB-reinforced concrete cylinders. It appears more significant for the cylinder with a relatively strong confinement.

• The approaching method, which applies an actual compressive strength of the SFCB for considering its compressive contribution in concrete cylinders (in Method III), gave the most accurate prediction for the load-carrying capacity of SFCB-reinforced concrete cylinder. However, its applicability and accuracy need to be verified with more experimental data in the future.

6) The number of self-citation in reference list is relatively high.

Response:

Thank you for indicating this issue. This paper is a follow-up study on the compressive behavior of SFCBs, and many of its experimental findings and conclusions are compared to previous works, which results in a relatively high self-citation rate.

Please see the attachment for detailed responses.

Thanks again for the reviewer’s comments and suggestions for improving the quality of this article. 

Reviewer 2 Report

This paper presents the results of an experimental study on the compressive performance of longitudinal steel-fiber-reinforced polymer composite bars in concrete cylinders confined by different composites.

The study investigated three types of different cylinders and different transverse confinements. Different equations were used to predict the load carrying capacity.

The topic is fit to the field of the journal. The paper is well structured and elaborated, however, some remarks could be taken into consideration.

Firstly, in the title “FPR” terminology is used, as well as in some other places in the text. As this acronym is nowhere introduced, this reviewer tends to believe that it is a typo of FRP. If this is correct this reviewer strongly believes if a paper is prepared with care cases like this should not happen.

Authors are invited to study more about FRPs and their important properties (e.g.: https://doi.org/10.1016/j.conbuildmat.2021.124193) to avoid such possible typos.

“The height and diameter of the cylinders were 300 mm and 150 mm, respectively.” Could authors comment, whether these dimensions are representative for columns or not. Furthermore, height of the cylinders should be included as well in figures.

Fig. 11. The title is not totally clear. Equations have no ratios. The property which is defined by the equation can have a predicted-to-experimental ratio.

Table 4. The title is not totally clear. Equations have no ratios. The property which is defined by the equation can have a predicted-to-experimental ratio.

Conclusions are collecting the main findings.

some minor typos, and proof-check

Author Response

1) Firstly, in the title “FPR” terminology is used, as well as in some other places in the text. As this acronym is nowhere introduced, this reviewer tends to believe that it is a typo of FRP. If this is correct this reviewer strongly believes if a paper is prepared with care cases like this should not happen.

Response:

Thank you so much for indicating this issue. Authors sincerely apologize for the misunderstanding caused by the typo. The correct abbreviation is "FRP." We have carefully reviewed the revision and corrected the typo throughout the paper.

2) Authors are invited to study more about FRPs and their important properties (e.g.: https://doi.org/10.1016/j.conbuildmat.2021.124193) to avoid such possible typos.

Response:

Thank you for your suggestions. We have benefited greatly from the reference and have cited it in the paper as follows (Ref. 13):

13. Solyom S, Di Benedetti M, Balázs GL. Bond of FRP bars in air-entrained concrete: Experimental and statistical study. Construction and Building Materials. 2021 Sep 20;300:124193.

3) The height and diameter of the cylinders were 300 mm and 150 mm, respectively.” Could authors comment, whether these dimensions are representative for columns or not. Furthermore, height of the cylinders should be included as well in figures.

Response:

Thank you for the comment. It is a misleading expression here. The concrete cylinder is 300 mm high and has a cross-sectional diameter of 150 mm. A revision has been made to clarify this issue as in L86-87 and as follow. Also, the height of the cylinders has been added into Figure 4(a), as follow.

The concrete cylinder is 300 mm high and has a cross-sectional diameter of 150 mm.

(a)

Figure 4. Test setup: (a) Test instruments; (b) Strain gauge distribution in SF-CF/SF-CD cylinder; (c) Strain gauge distribution in SF-BS cylinder.

4) Fig. 11. The title is not totally clear. Equations have no ratios. The property which is defined by the equation can have a predicted-to-experimental ratio.

Response:

Thank you for indicating this issue. The title of Figure 11 has been revised as follow:

Figure 11. Predicted-to-experimental ratios of the design equations for predicting the test cylinders.

5) Table 4. The title is not totally clear. Equations have no ratios. The property which is defined by the equation can have a predicted-to-experimental ratio.

Response:

Thank you for indicating this issue. The title of Table 4 has been revised as follow:

Table 4. Predicted-to-experimental ratio results of the design equations for predicting the test cylinders.

6) Conclusions are collecting the main findings.

Response:

Thanks for the comment. The conclusion has been simplipied and restructured as follows:

• The post-yield stiffness of the confined SFCB developed relatively higher when it was confined with a relatively high elastic modulus CFRP composite than when it was confined with a relatively low elastic modulus of BFRP composite.

• The compressive failure strain of SFCB in SF-CF cylinder could have reached 88% of its tensile peak strain, indicates that a relatively high utilization of the material strength may be achieved for the SFCB restrained with relatively high elastic modulus confinements.

• The design equations that consider the compressive contribution of SFCBs in concrete cylinders with a strength reduction factor of SFCB (in Method I) or a maximum compressive strain of concrete (in Method â…¡) generally underestimated the load-carrying capacity of SFCB-reinforced concrete cylinders. It appears more significant for the cylinder with a relatively strong confinement.

• The approaching method, which applies an actual compressive strength of the SFCB for considering its compressive contribution in concrete cylinders (in Method III), gave the most accurate prediction for the load-carrying capacity of SFCB-reinforced concrete cylinder. However, its applicability and accuracy need to be verified with more experimental data in the future.

Please see the attachment for detailed responses.

Thanks again for the reviewer’s comments and suggestions for improving the quality of this article. 

Round 2

Reviewer 1 Report

The paper is recommended for publication. 

The English of paper is good.